# Pivotal Role of Intestinal Microbiota and Intraluminal Metabolites for the Maintenance of Gut–Bone Physiology

**DOI:** 10.3390/ijms24065161

**Published:** 2023-03-08

**Authors:** Niklas Grüner, Anna Lisa Ortlepp, Jochen Mattner

**Affiliations:** 1Mikrobiologisches Institut—Klinische Mikrobiologie, Immunologie und Hygiene, Universitätsklinikum Erlangen and Friedrich-Alexander Universität (FAU) Erlangen-Nürnberg, 91054 Erlangen, Germany; 2Medical Immunology Campus Erlangen, Friedrich-Alexander Universität (FAU) Erlangen-Nürnberg, 91054 Erlangen, Germany

**Keywords:** gut–bone axis, intestinal microbiota, intestinal dysbiosis, hormone and electrolyte metabolism, cell trafficking, inflammatory bowel disease, cytokines, arthritis, osteoporosis, spondyloarthropathy

## Abstract

Intestinal microbiota, and their mutual interactions with host tissues, are pivotal for the maintenance of organ physiology. Indeed, intraluminal signals influence adjacent and even distal tissues. Consequently, disruptions in the composition or functions of microbiota and subsequent altered host–microbiota interactions disturb the homeostasis of multiple organ systems, including the bone. Thus, gut microbiota can influence bone mass and physiology, as well as postnatal skeletal evolution. Alterations in nutrient or electrolyte absorption, metabolism, or immune functions, due to the translocation of microbial antigens or metabolites across intestinal barriers, affect bone tissues, as well. Intestinal microbiota can directly and indirectly alter bone density and bone remodeling. Intestinal dysbiosis and a subsequently disturbed gut–bone axis are characteristic for patients with inflammatory bowel disease (IBD) who suffer from various intestinal symptoms and multiple bone-related complications, such as arthritis or osteoporosis. Immune cells affecting the joints are presumably even primed in the gut. Furthermore, intestinal dysbiosis impairs hormone metabolism and electrolyte balance. On the other hand, less is known about the impact of bone metabolism on gut physiology. In this review, we summarized current knowledge of gut microbiota, metabolites and microbiota-primed immune cells in IBD and bone-related complications.

## 1. Introduction

The gastrointestinal tract hosts a broad range of highly diverse microorganisms, including viruses, parasites, fungi and bacteria. All of these microorganisms are referred to as intestinal microbiota. The number of intestinal microbiota might even outnumber the body’s own cells [1,2,3]. The interactions between commensal microbiota and the host’s own cells in the gut are pivotal for many physiological organ functions and the maintenance of the body’s immune homeostasis. Therefore, the composition of microbial species in the gut, and their combinational interactions with host tissues, are unique for each individual. Dietary, environmental and genetic factors of the host influence the composition and function of intestinal microbiota [4,5,6,7]. However, genes of different microbiota share overlapping and redundant functions [8,9], so that compositional changes of gut microbiota do not necessarily reflect metabolic and/or functional alterations. 

Microbiota play pivotal functions in the physiology of the host. They help to digest dietary products, contribute to the production of vitamins and produce—frequently in mutual concert with the host—various metabolites that play essential roles for the maintenance of organ physiology [10,11,12,13]. Moreover, intestinal microbiota are pivotal for the development of gut-associated lymphoid tissues and the induction of local and systemic immune responses [14,15]. Resident microbiota also regulate the development and differentiation of intestinal immune cell populations and shape the local cytokine milieu in the gut [16]. Microbiota of the gut are not only important for the development of gut-associated lymphoid tissue and the generation of certain immune cells, but also for the maintenance or restoration of non-inflammatory homeostasis in the intestine, attributable in part to microbial metabolites [16,17,18,19]. Additionally, the microenvironment of the tissue, as well as spatial and/or temporal interactions between the immune system of the host and the intestinal microbiota, shape the local immune response [20]. This latter could even exhibit situationally opposite effects. Various cytokines and immune cells also influence the composition and function of the intestinal microbiota. However, the mechanisms by which specific bacterial populations induce the development of individual cell subsets, or by which the cytokine milieu shapes the composition of the intestinal microbiota, remain poorly understood. 

Microbiota and their metabolites influence the local tissue environment of the gut. They also signal to distant tissues, such as the liver, the bone or the brain [12,21,22]. These physiological interactions can change significantly when the composition of the microbiota and/or the permeability of the intestinal epithelial barrier changes. Consequently, many disorders have been associated with intestinal dysbiosis. Indeed, leaky gut syndrome is associated with many inflammatory and immune-mediated diseases, including inflammatory bowel disease (IBD), type 1 diabetes (T1D), osteoporosis and arthritis [23,24,25,26,27]. Interestingly, although intestinal barrier dysfunction contributes to chronic, low-grade inflammation during aging, it has yet to be determined whether restoring barrier function is able to ameliorate clinical manifestations in gastrointestinal or systemic diseases [28,29]. Particularly, the role of the intestinal barrier in the gut–bone axis, under physiological and pathophysiological conditions, has become a focus of recent interest [24,28].

## 2. The Contribution of the Gut–Bone Axis to Bone Metabolism

Interactions between the gut and bones are complex. Intestinal microbiota, for example, can interfere with various signaling circuits that regulate bone metabolism (Figure 1). We summarized, herein, some of the most important electrolytic, hormonal and metabolic factors that interfere with bone homeostasis, and described the pathways that microbiota influence. Electrolytes, like calcium and magnesium, or minerals like iron are pivotal for bone mineralization. Calcium intake influences skeletal calcium retention during growth and thus affects peak bone mass in early adulthood [30]. Moreover, magnesium intake is associated with higher hip and femoral neck bone mineral density [31]. Iron is essential for bone metabolism and has been positively related to bone mineral density [32]. On the other hand, disruptions of iron metabolism, leading to both iron overload and iron deficiency, are associated with low bone mineral density and fragility [33]. Iron deficiency anemia is another common systemic manifestation of IBD [34,35]. Microbiota also interfere with electrolyte and mineral metabolism. Thus, for example, specific intestinal microbiota can bind iron in the large intestine, limiting free radical formation. Certain probiotics affect calcium and phosphate resorption, as well, due to the regulation of the small intestinal barrier [36]. 

Vitamin D aids in the intestinal absorption of calcium. As it is synthesized in the skin or absorbed from the diet, vitamin D is converted, over several steps, into its active form. The active form, 1,25-dihydroxy vitamin D, subsequently interacts with the vitamin D receptor (VDR) to modulate the expression of target genes [37]. Thus, 1,25-dihydroxy vitamin D regulates bone and cartilage anabolism and the availability of calcium and iron for physiological tissue matrix mineralization [37,38]. Vitamin D supplementation increases the diversity of the gut microbiota, and vitamin D deficiency is associated with gut dysbiosis and inflammation [39]. 

Next to 1,25-dihydroxy vitamin D, there are two other hormones that play significant roles in electrolyte metabolism in the body, as well as the regulation of bone remodeling and density. These are parathyroid hormone (PTH) and calcitonin. Both are essential for the maintenance of mineral bone physiology, and both regulate calcium–phosphate metabolism. Low calcium levels trigger PTH production. Low and intermittent doses of PTH exhibit anabolic effects on bone volume and microarchitecture. In contrast, a continued hypersecretion of PTH, such as that which occurs in primary hyperparathyroidism, leads to bone resorption [40,41]. Acting as an opponent of PTH, calcitonin reduces calcium concentrations in the blood, for example, due to the inhibition of bone suppression or the suppression of calcium release from the bone [42]. Thus, calcitonin can retain bone density and reduce the risk of fractures. Both calcitonin and PTH require elements of the intestinal microbiota to exert catabolic and anabolic effects on bone metabolism. For example, many gut microbiota are able to digest carbohydrates to generate short chain fatty acids (SCFAs). These promote the differentiation of regulatory T cells (Tregs) from naive T lymphocytes [43,44,45]. Tregs mediate the bone anabolic activity of the probiotic *Lactobacillus rhamnosus* [46,47]. Thus, as Tregs interfere with PTH-induced bone metabolism [48], Treg differentiation is pivotal for the bone anabolic activity of PTH [49]. 

Another hormone that contributes to the growth and maturation of bones is estrogen. This steroid hormone regulates mammalian reproduction bone turnover in adult bones and promotes the proper closure of the epiphyseal growth plates during bone growth [50]. Decreased estrogen production in postmenopausal women is an underlying factor in the rapid bone resorption, resulting in osteoporosis, that many women experience [51,52]. Furthermore, estrogen deficiency can trigger the release of TNF-alpha, leading indirectly to bone loss and osteoclastogenesis [53]. The gut microbiota secretes an enzyme, beta-glucuronidase, which deconjugates estrogens into their active forms [54]. Thus, intestinal microbiota regulate the activity and functions of estrogen. 

The mutual crosstalk of the host and the intestinal microbiota also extends to other factors that are pivotal to bone homeostasis and skeletal microarchitecture [55]. Typically, microbiota promote catabolic effects on bone homeostasis [56] and regulate physiological bone turnover [57]. Furthermore, microbiota shape the local cytokine milieu and can affect bone metabolism via the release of cytokines. Particularly, the balance between Th17 and regulatory T lymphocytes (Tregs) is critical for bone metabolism [55]. Short chain fatty acids (SCFAs), released by microbiota upon carbohydrate digestion, are also important modulators of bone physiology [45,58,59,60]. We discussed the consequences of disruption in Th17 and Treg homeostasis and reduced SCFA production on bone physiology in more detail in Section 6.2 and Section 6.5. Serotonin and its precursor tryptophan are both primarily synthesized in the gut, and are therefore influenced by intestinal microbiota. They can regulate bone mass, and can also exhibit antagonistic functions, based on the body compartment in which they are acting [61].

Biliary acids mediate other secondary effects of microbiota. For example, following engagement of the farnesoid X receptor (FXR) and the G protein-coupled bile acid receptor 5 (TGR5), intestinal microbiota can change the amount and type of secondary bile acids [62]. TGR5 ligation by secondary bile acids can induce the production of glucagon-like peptide-1 (GLP-1), which in turn activates the proliferation of thyroid C cells and the secretion of calcitonin, thus, inhibiting bone resorption [63]. GLP-1 can also trigger the proliferation of osteoblasts and inhibit osteoclast activity [64]. Furthermore, monohydroxylated secondary lithocholic acid (LCA) can serve as a direct ligand for the vitamin D receptor [65]. Thus, intestinal microbiota are crucial for directly and indirectly regulating the gut–bone axis. Consequently, intestinal dysbiosis can disrupt bone physiology. One of the pathological conditions that can simultaneously affect the gut and the bone is inflammatory bowel disease (IBD). Therefore, we discussed osteoporosis and arthritis as separate disorders—and as two common extraintestinal disease manifestations of IBD patients.

## 3. Compositional Changes of the Intestinal Microbiota as Signature of IBD

Epithelial surfaces, such as the gut, form a large interface between the body and the external environment. An immune-mediated inflammation of the gastrointestinal (GI) epithelium characterizes inflammatory bowel diseases (IBD), such as Crohn´s disease (CD) or ulcerative colitis (UC) [66,67]. Clinical relapses, as well as severe gastrointestinal mucosal and transmural lesions in a non-sterile environment, characterize both disorders [68,69]. Although the exact etiologies underlying IBD remain elusive, these disorders presumably result from a complex interplay of microbial, genetic, geographic and habitual factors, which consequently disrupts interactions of intestinal microbiota, the intestinal epithelium and the immune system [70,71,72]. Various cellular, cytokine and molecular pathways perpetuate immune dysfunction in IBD [71,73,74]. Biologicals targeting these pathways, at best, induce remission in just half of the patients [75,76,77]. 

Intestinal microbiota are critical components in the pathogenesis of IBD, as they influence nutrient metabolism and host immune responses. IBD patients exhibit an altered composition of gut microbiota compared to healthy individuals [78,79]. Increasing evidence implies that the microbiota initiate and maintain intestinal inflammation [80]. Common pathogenic mechanisms include an increased exposure of bacterial antigens to intestinal immune cells of the host and/or alterations in the immune response of the host to commensal bacteria [81,82,83,84,85]. Whether these alterations in the composition of the intestinal microbiota are the consequence, or the cause, of intestinal inflammation has become a subject of intensive investigation. Fecal microbiota transfers (FMTs) from healthy donors, replacing dysbiotic microbiota in IBD recipients, could be a novel treatment for IBD [17,18,19,86,87,88,89]. 

## 4. The Bone as Frequent Target of Extraintestinal IBD Manifestations

Diarrhea, rectal bleeding, abdominal pain, fatigue and weight loss accompany both UC and CD [72,90,91]. Typically, periods of active disease follow periods of remission. Depending on the severity of inflammation and the affected parts of the gut, symptoms vary and may range from mild to severe. Furthermore, IBD patients frequently experience extraintestinal manifestations [92,93] and complain about musculoskeletal, ocular and/or cutaneous symptoms in addition to gastrointestinal symptoms. Commonly, these extraintestinal symptoms (which occur in 5% to 50% of all IBD patients) result in significant morbidity, even more than intestinal disease itself [94]. Impacts on the musculoskeletal system are among the most common extraintestinal manifestations; osteoporosis and arthritis are two common clinical problems accompanying IBD [94,95] (Figure 2). 

Enteropathic arthritis, associated with IBD, is distinct from rheumatoid arthritis; it belongs to an extended disease group of inflammatory arthritides, namely the spondyloarthropathies (SpA). These share common clinical features, including axial spondylitis, sacroiliitis, peripheral inflammatory arthritis, enthesitis and dactylitis [95,96,97,98,99,100]. Additional extraintestinal manifestations of IBD affecting bone and joints include ankylosing spondylitis, avascular hip necrosis and osteomalacia [94] (Figure 2). Although most extraintestinal manifestations, including pauci-articular arthritis, are directly associated with ongoing intestinal disease flares, others, like ankylosing spondylitis, occur independently of intestinal disease activity [94].

### 4.1. Osteoporosis/Osteopenia 

Bone is a living tissue that undergoes constant remodeling by bone-forming cells (osteoblasts) and bone-resorbing cells (osteoclasts). Imbalances in osteoblastic bone formation and osteoclastic bone resorption lead to osteopenia or osteoporosis [101,102,103,104]. Both osteopenia and osteoporosis are quantitative, rather than qualitative, metabolic disorders of bone mineralization [105]. 

A decrease in bone mineral density (BMD) characterizes both disorders. Osteopenia shows BMD scores below normal reference values, but not low enough to meet the diagnostic criteria for osteoporosis [104]. Thus, the extent of bone microarchitecture disruption makes it possible to distinguish one disease from the other [104,105]. The more pronounced the bone loss, the more fragile the bone becomes, resulting in increased risk of fracture and disability [103].

Both osteoporosis and osteopenia are multifactorial diseases [103]. Endocrine mechanisms, such as estrogen or vitamin D deficiency, as well as secondary hyperparathyroidism, underlie their pathogeneses, similarly to dysregulation in interactions between bone and the immune system, dysbiosis of the gut microbiota and cellular senescence [103,104,105]. Indeed, some of the same pathogenic mechanisms underlie both IBD and osteopenia/osteoporosis. 

### 4.2. Influence of Intestinal Microbiota on Postmenopausal Osteoporosis

Osteopenia and osteoporosis are common metabolic bone diseases in postmenopausal women. Postmenopausal cessation of ovarian function—and the consequent estrogen deficiency—is the primary cause of both metabolic bone diseases (Figure 1). Declining estrogen levels result in the stimulation of bone resorption and—to a lesser extent—bone formation, leading to a period of rapid bone loss [106].

A growing body of evidence suggests that the gut microbiota are involved in the regulation of bone metabolism [63,107,108]. In comparisons of healthy postmenopausal women with women suffering from osteopenia and osteoporosis, the composition of microbiota significantly differed. While the intestinal microbiota of healthy controls contained more *Clostridia* and *Methanobacteriaceae*, more *Bacteroides* were recovered from the feces of osteopenic and osteoporotic women [109]. Another study confirmed alterations in the composition of intestinal microbiota in postmenopausal women. The authors observed an accumulation of *Fusicatenibacter*, *Lachnoclostridium*, and *Megamonas* spp. in women with osteoporosis, along with an increase of TNF-alpha serum levels and a decrease of serum IL-10 concentrations. Although they observed additional alterations in the composition of vaginal microbiota, they concluded that compositional changes in the intestinal microbiota were more closely correlated to osteoporosis [110]. 

In mice, ovariectomy (ovx) or treatment with gonadotropin-releasing hormone (GnRH) agonists modeled the effects of estrogen depletion [111,112]. Microbiota promoted osteoporosis in sexual-hormone-deficient mice. However, dietary supplementation of specific probiotics, such as *Lactobacillus rhamnosus*, protected mice from osteoporosis by strengthening the barrier integrity of the gut and, consequently, alleviating inflammation [113]. Accordingly, fecal microbiota transplants (FMTs) protected recipients from osteoporosis as well. A reduced production of osteoclastic cytokines, such as TNF-alpha or IL-1beta, and an increase in SCFAs accompanied that protection [114]. Moreover, FMTs strengthened the intestinal barrier, as demonstrated by increased expression of tight junction proteins. 

### 4.3. Osteoporosis in IBD

Several gastrointestinal disorders have been associated with osteoporosis and osteopenia, including IBD, celiac disease and chronic liver disease [115,116,117,118]. Indeed, osteoporosis and arthritis are leading causes of morbidity in IBD patients [94]. Bone loss is an early systemic process and can occur even before clinical disease manifests. Thus, extraintestinal symptoms affecting the bone need to be considered in IBD therapy.

IBD has been associated with decreased bone mass and alterations in bone geometry from the time of diagnosis, frequently before the initiation of anti-inflammatory therapy. Bone disease is attributed to vitamin D deficiency, steroid use and/or systemic inflammation [119]. IBD patients are at higher risk for developing osteoporosis and osteopenia than the general population, with a 40% higher relative risk of fracture in IBD patients [120,121,122]. The prevalence of osteopenia and osteoporosis in IBD patients varies significantly depending on the study populations, location and design, ranging from 22% to 77% and from 17% to 41%, respectively [122,123,124]. 

The etiology of osteoporosis in IBD is multifactorial, with risk factors including age, (long-time) corticosteroid use, (protein-calorie) malnutrition, vitamin D and calcium malabsorption and deficiency, immobilization and the underlying inflammatory state [120,121,122,125,126]. Inactivity, hypogonadism and stunted growth in children, as well as decreased skeletal muscle mass, also most likely play a role in the pathogenesis. Deficits in bone mass can persist despite the absence of symptoms of active IBD.

The effects of IBD on the skeleton are complex. Preliminary studies suggested that the dysbiotic intestinal microbial flora present in IBD could also affect bone at a distance. Several mechanisms underlying dysregulated gut–bone interactions are possible. For example, T cells activated by the gut microbiota may serve as inflammatory shuttles between the intestine and the bone (Figure 3). Microbe-associated molecular patterns leaked into the circulation in IBD could activate immune responses in the bone marrow by immune cells, osteocytes, osteoblasts and osteoclasts, leading to decreased bone formation and increased resorption. Finally, intestinal microbial metabolites, such as hydrogen sulfide (H_2_S), may also impair bone cell functions. Uncovering these mechanisms will enable the design of microbial cocktails to help restore bone mass in IBD patients [127]. 

Accordingly, in almost 15% of IBD patients that were analyzed in a prospective, single center study, markers for bone resorption, such as the C-terminal telopeptide of type 1 collagen (CTX), were increased. Bone mineral alterations were common in IBD patients and Vitamin D supplementation was found to be crucial, especially when taking corticosteroids, azathioprine and/or infliximab [119]. Interestingly, the severity (but not the activity) of disease was associated with osteopenia in IBD patients [128,129,130]. Furthermore, patients suffering from IBD in their younger years had a lower bone density compared to healthy controls [131,132]. Moreover, IBD patients are at increased risk for fractures. Particularly, young IBD patients with high inflammatory burdens or corticosteroid exposure may suffer from bone mineral loss and exhibit a higher risk of fractures in later years. An earlier, more aggressive therapy to reduce inflammation and/or the use of steroid-sparing drugs could reduce the risk. Furthermore, IBD patients should have their bone density checked frequently [133].

### 4.4. Arthritis and Spondyloarthropathies in IBD

Significant numbers of patients with spondyloarthropathies (SpA) suffer from associated clinical IBD. Nearly half of them show subclinical gut inflammation; however, the connection between the gut and the musculoskeletal system has remained an exasperating problem [96]. Musculoskeletal symptoms include the most common extraintestinal manifestations of IBD. Peripheral arthritis is the most common extraintestinal manifestation in CD and UC patients [134]. An observational two-year cross-sectional study reported that 30% of IBD patients with musculoskeletal complaints for more than three months had SpA. SpA occurred more frequently in men and patients with surgery for IBD [135]. They accounted for about 40% of all extraintestinal manifestations and manifest mainly as axial or peripheral spondyloarthritis. Spondylitis ankylosans was found in 5–10% of IBD patients. About 25% of IBD patients suffered from sacroiliitis [94] (Figure 2).

Another study confirmed arthritis as the most common extraintestinal complication of CD and UC. Indeed, more than 21% of CD patients and 12.5% of UC patients suffered from arthritis, suggesting that arthritis was even more frequent in CD than in UC [136]. Prevalence of extraintestinal manifestations in the locomotor system decreased with age, with a prevalence of almost 25% in 20–30 year old patients, down to 2% in 50–60 year old patients. Peripheral arthritis occurred in 5–14% of UC patients and in 10–20% of CD patients [93]. 

## 5. Animal Models of Inflammatory Bone Loss 

The close relationship between inflammation and bone metabolism has been appreciated and recognized for a long time [66,81,137]. Toll-like receptors (TLRs) play pivotal roles in inflammation and provide important links between the immune and skeletal systems. In a study, TLR9^−/−^ mice exhibited low bone mass and low-grade systemic chronic inflammation, which as characterized by the expansion of CD4^+^ T cells and increased levels of inflammatory cytokines, including TNF-alpha, RANKL (in detail described below) and IL-1beta. The increased levels of these cytokines significantly promoted osteoclastogenesis and induced bone loss. Importantly, TLR9 deletion altered the gut microbiota, and this dysbiosis was an underlying factor in the systemic inflammation and bone loss observed in TLR9^−/−^ mice [138]. 

Short chain fatty acids (SCFAs ) are microbiota-derived metabolites that can potentially influence bone homeostasis. They mainly consist of acetate, butyrate and propionate. Indeed, these products of dietary fibers enhance alkaline phosphatase activity, a marker of osteoblast differentiation, in cell lines in vitro. Thus, SCFAs might contribute to the maintenance of a positive balance of bone turnover [58]. SCFAs also regulate osteoclast metabolism and bone mass in vivo. Treatment of mice with SCFAs or feeding with a high-fiber diet significantly increased bone mass and prevented postmenopausal and inflammation-induced bone loss. The protective effects of SCFAs on bone mass were primarily associated with inhibition of osteoclast differentiation and bone resorption, while bone formation was less affected [59]. 

The Winnie mouse, which carries a mutation in the *Muc2* gene and serves as a model of spontaneous chronic colitis, also exhibited colitis-associated bone loss under study. Compared to B6 controls, these defects included a deterioration in trabecular and cortical bone microarchitecture, increased bone resorption, and decreased bone formation and bone strength. Moreover, the number of osteoblasts decreased, while the number of osteoclasts increased. The onset and progression of intestinal inflammation were associated with increased gut-derived serotonin levels. Thus, the skeletal phenotype of Winnie mice closely resembled the clinical manifestations of IBD-associated osteoporosis/osteopenia [139]. 

## 6. Dysregulated Gut–Bone Interactions Potentially Underlying Skeletal Disease Manifestations in IBD

As outlined above, the bone is a frequent target of extraintestinal IBD manifestations. Multiple mechanisms might underlie pathological alterations in the gut–bone axis and the subsequent induction of bone disease. These include intestinal dysbiosis, disrupted cytokine and chemokine responses, a deterioration of intestinal barrier integrity, aberrant vessel formation and metabolic alterations. Below, we discussed several of these in greater detail. 

### 6.1. Intestinal Dysbiosis

By comparing microbiota in healthy individuals with IBD patients, patients with ankylosing spondylitis (AS) or patients suffering from both, it was noted that the composition of microbiota from AS and IBD patients differed from healthy individuals [140]. Moreover, alpha- and beta-diversity were altered. *Streptococcus* and *Haemophilus* accumulated in both IBD and AS patients. However, the authors noted that microbiota might not have been underlying the pathogenesis of either disease, because microbiome composition varied along with different geographic regions independently of the disease status. This indicated that other factors could introduce bias into studies of the associations between microbiome composition and disease activity. Another study also detected alterations in the composition of microbiota. Therein, *Clostridiaceae* increased among the intestinal microbiota of patients with IBD and rheumatoid arthritis (RA) [141]. In another study, the authors speculated that molecular similarities between antigenic epitopes of gut microbiota and host tissues could possibly have induced cross-reactivity of T cells. Moreover, the increased epithelial barrier permeability could have been an underlying factor in the extraintestinal manifestations of IBD and systemic inflammatory reactions. Dysbiosis could therefore have activated intestinal immune cells and induced their migration to other organs outside the gut [93]. However, the specific microbiota underlying these pathogenic effects were difficult to define. Although another study confirmed alterations in the composition of intestinal microbiota, they observed an accumulation of other bacterial species. The numbers of *Actinomyces*, *Eggerthella*, certain *Clostridia*, *Faecalicoccus* and *Streptococcus* significantly increased in immune-mediated inflammatory diseases. CD patients exhibited the greatest variability in the composition of microbiota [142]. Consequently, there were microbiota that accumulated in both IBD patients and patients with chronic rheumatic diseases, such as *Bifidobacterium*, *Staphylococcus*, *Enterococcus*, *Lactobacillus*, *Pseudomonas*, *Klebsiella* and *Proteus*. In contrast, *Faecalibacterium* and *Roseburia* decreased in both disorders. However, some bacteria, such as *Eubacterium*, *Clostridium*, *Ruminococcus* and *Coprococcus*, specifically increased in chronic rheumatic disease, but decrease in IBD [142]. 

### 6.2. Cytokines

Antigens derived from gut microbiota are key targets for intestinal effector T cell activation and differentiation [143,144]. Particularly, the balance between T helper 17 (Th17) and regulatory T lymphocytes (Tregs) is critical for bone metabolism [55]. Indeed, disruptions of this Th17-Treg balance promote both intestinal and joint inflammation [145], and the Th17 pathway has been linked to both intestinal and joint disease [145]. Dysregulated microbiota drive Th17 cell expansion and immune cell migration to the joints [96,146,147,148] (Figure 3). Segmented filamentous bacteria (SFB), for example, enable parathyroid hormone (PTH) to expand intestinal TNF-alpha- and IL-17-producing T cells and promote their egress from the gut. Following migration into the bone marrow, Th17 cells induce the production of the receptor activator of nuclear factor kappa-Β ligand RANKL by osteoblasts and osteocytes, causing loss of bone mass [149]. RANKL controls bone regeneration and remodeling. Following stimulation with TNF-alpha or IL-17, a variety of different cell populations, including T- and B-lymphocytes and osteoblasts, release this primary driver of osteoclastogenesis [53,150,151,152]. Macrophage colony-stimulating factor (M-CSF) is the cytokine that initiates early osteoclast differentiation [72]. Vitamin D receptor negatively regulates bacterial-stimulated NF-kappa B activity in the intestine, demonstrating a reciprocal interaction between calcium homeostasis and microbiota [153]. 

Thus, alterations of the microbiota might change the antigen pattern and, consequently, the cytokine response. Inflammatory mediators such as IL-6, TNF-alpha, IFN-gamma and vascular endothelial growth factor (VEGF) are elevated in the serum of IBD patients. These systemic inflammatory responses can activate the production of cytokines in non-intestinal organs [92].

### 6.3. Intestinal Barrier Integrity and Immune Cell Trafficking

An impaired intestinal barrier precedes clinical diagnosis of IBD by years [154,155]. Intestinal microbiota and diurnal variations in diet microbiota interactions might be regulators of the intestinal barrier [156]. Under study, IBD patients suffering from concomitant spondyloarthritis exhibited enhanced epithelial permeability in the ileum. Expression of the tight junction proteins occludin and claudin-1/-4 was lowered, but there was an increase in bacterial infiltrates in the gut wall. Furthermore, these patients had enhanced levels of serum proteins such as I-FABP, LPS and sCD14 in their sera, indicating interactions with intestinal microbiota [157]. 

Along with the findings regarding altered intestinal permeability, a number of unique lymphocyte populations expand within the gut and the skin of patients with SpA, including gamma/delta T cells, mucosa-associated invariant T (MAIT) cells, innate lymphoid cells (ILCs) and T resident memory (TRM) cells. These cells respond to microbial cues at their barrier surface, causing cellular activation and generation of interleukin (IL)-17, which is hypothesized to be the mechanism by which the cells contribute to SpA pathogenesis [158]. Using photolabeling gut–joint trafficking of intraepithelial lymphocytes (IELs) to joint enthesis, the pathogenic site of SpA has been visualized [159]. Moreover, different adhesion molecules that mediate immune cells, homing in on the gut and joints, have been uncovered. The binding of intestinal lymphocytes to inflamed synovium depends on vascular adhesion protein-1 (VAP-1)—but not alpha4 beta7–MAdCAM-1—interactions [160]. The adherence of macrophages is mainly P-selectin dependent [161]. Gut-derived mucosal immune cells of IBD patients can bind to inflamed synovial venules, as well. Lymphocytes, for example, use VAP-1, CD44, CD18 integrins, intracellular adhesion molecule-1 (ICAM-1), L-selectin and peripheral lymph node addressins (PNAd), as well as interactions between alpha4 integrins and vascular cell adhesion molecule-1 (VCAM-1), to bind to synovial tissues. Mucosal macrophages, on the other hand, utilize P-selectin, and its ligand, P-selectin glycoprotein ligand-1 (PSGL-1), as well as E-selectin and VAP-1, for interactions with the synovium [162].

### 6.4. Angiogenesis

Gut microbiota can also affect the vascular system. Patients with acute coronary syndrome, for example, exhibited distinct serum metabolome and gut microbial signatures, as compared to control individuals [163]. Even in healthy individuals, lower gut microbial diversity was associated with higher white blood cell counts and C-reactive protein levels [164]. Moreover, patients with Behcet´s disease, a systemic autoimmune inflammation of the blood vessels, exhibited a different composition of intestinal microbiota, compared with healthy adults [165]. Gut dysbiosis, with elevated production of IL-6, IL-1beta, TNF-alpha and VEGF exacerbated pathological angiogenesis [166]. 

Similar microbiota-dependent effects likely play a role in the pathogenesis of IBD, as the intestinal vascular endothelium builds a second pivotal barrier next to the intestinal epithelium in the gut [74,167]. Gut microbiota can facilitate vascular dysfunction and hypertension, at least in part, due to MCP-1-/IL-17-driven vascular immune cell infiltration and inflammation [168]. Furthermore, inflammation activates angiogenesis in IBD [74,169,170,171]; dysfunction of the vascular barrier represents another important pathomechanism of IBD [74]. Microbiota contribute to the maintenance of vascular tone via actin polymerization [172]. Moreover, microbiota influence the enteric nervous system [173], which regulates intestinal permeability and blood vessels [174,175,176,177,178]. 

Similarly to IBD, arthritis is an angiogenesis-dependent disease. Angiogenesis is an early event in the inflammatory joint. It allows activated immune cells to enter the synovium [179]. Moreover, vascular turnover increases in the arthritic joint [55]. Increased vascular proliferation and/or blood flow could contribute to gut–joint trafficking of immune cells. Thus, the inflamed intestinal vasculature could provide novel therapeutic targets for the treatment of intestinal and extraintestinal IBD symptoms.

### 6.5. Metabolic Alterations

Similarly to what has been observed in osteoporosis, SCFAs protect against collagen-induced arthritis. In an experimental model of rheumatoid arthritis (RA), free fatty acid receptor 2 (FFA2) on CD19+ B cells underlay the protective effects [60]. TNF-alpha-overexpressing mice spontaneously developed IBD and RA, confirming the pivotal role of microbiota and microbiota-derived products in the pathogenesis of both diseases. FMTs of TNF-alpha-overexpressing donors into germ-free mice also induced IBD and RA in the recipients. The induction of autoreactive T cells due to dysbiotic gut microbiota could therefore represent a disease-mediating mechanism [180]. 

IBD and RA are also characterized by disrupted amino acid metabolism [96,181,182]. We briefly discussed this disrupted interaction with respect to the semi-essential amino acid L-arginine. For example, IBD patients exhibited altered availability of L-arginine in different tissue compartments [183,184,185]. Along with the reduced intraluminal L-arginine levels, a dietary L-arginine supplementation protected against DSS-, oxazolone- and *Citrobacter*-induced colitis in preclinical models [186,187,188,189]. Accordingly, an upregulation of the expression of the argininosuccinate lyase (ASL), the only enzyme able to produce L-arginine, was correlated with improved epithelial integrity and alleviation of colitis [190,191]. In contrast to the inflamed gut tissues, L-arginine concentrations increased in the plasma of IBD and RA patients, and could potentially mediate systemic adverse effects or disease [183,192]. Indeed, L-arginine might promote experimental arthritis, as signs of inflammation inversely correlated with the availability of L-arginine [192]. Accordingly, L-arginine deficiency, in multinucleated giant cells, improved symptoms of arthritis, as indicated by reduced swelling and weight loss [193]. In another study, dietary L-arginine application ameliorated experimental arthritis [194]. Consequently, the availability of L-arginine in distinct cellular and tissue compartments is critical for the influence of immune responses in both diseases. How this links IBD and RA together will be the subject of future investigations. 

Another example of altered metabolic gut–bone interactions is the gut microbiota-related metabolite trimethylamine N-oxide (TMAO). TMAO is a biologically active molecule, and is putatively involved in the promotion of different chronic diseases in humans [195]. In contrast, a reduction of plasma TMAO levels was associated with a greater loss of bone mineral density (BMD) in patients with type 2 diabetes [196]. Thus, as type 2 diabetes is related to obesity and altered bone health, by applying diets aiming for weight loss, thereapies must also consider adequately balancing TMAO and its precursors, choline and l-carnithine, as TMAO could prevent BMD reduction.

## 7. Conclusions

Intestinal microbiota and metabolites are pivotal for the maintenance of gut–bone physiology. They have been proven able to influence the absorption of calcium, magnesium and iron, to affect hormone metabolism and to directly and indirectly influence barrier integrity and immune responses. Specific probiotics, such as *Lactobacilli* or *Bifidobacteria*, regulate bone health [55,197]. Dysbiosis, on the other hand, leads to bone pathology in several ways—for example, reduced electrolyte absorption, altered metabolism, the expansion, egress and joint trafficking of Th17 cells, pathological angiogenesis and/or a disrupted barrier permeability. Clinical remission, endoscopic healing, absence of disability and a restoration of quality of life are the most important long-term achievable treatment targets [198]. These are most frequently achieved by potent anti-inflammatory drugs, e.g., antibodies like Adalimumab or Infliximab [92], or locally active or systemic therapies, with significant side effects [199]. Thus, alternative therapeutic approaches targeting intestinal and extraintestinal symptoms are urgently required. It would be intriguing to treat IBD by changing gut microbiota and intraluminal metabolites using simple dietary supplements. For instance, probiotics might induce remission in UC patients for a limited time [200]. Another method to preserve or reconstitute the gut microbiome is fecal microbiota transplantation (FMT). This has been well-established, and has demonstrated a very high degree of efficacy in patients suffering from refractory *Clostridioides difficile* infection (CDI). It also has a very good safety profile [201]. FMT has been studied in mice with experimental UC, where it improved Th1/Th2 and Th17/Treg imbalances through the regulation of intestinal microbiota [202]—which also affected musculoskeletal manifestations of IBD. In UC patients, FMT has shown promise, significantly improving rates of clinical remission in a recent meta-analysis [203]. Thus, it will be interesting to explore whether FMTs could also cure extraintestinal manifestations. Further research will hopefully clarify the mechanisms of inflammation-induced bone loss in IBD and guide effective treatment modalities [204]. In addition to FMTs, these could include the reconstitution of distinct microbial metabolites, such as SCFAs.

## Figures and Tables

**Figure 1 ijms-24-05161-f001:**
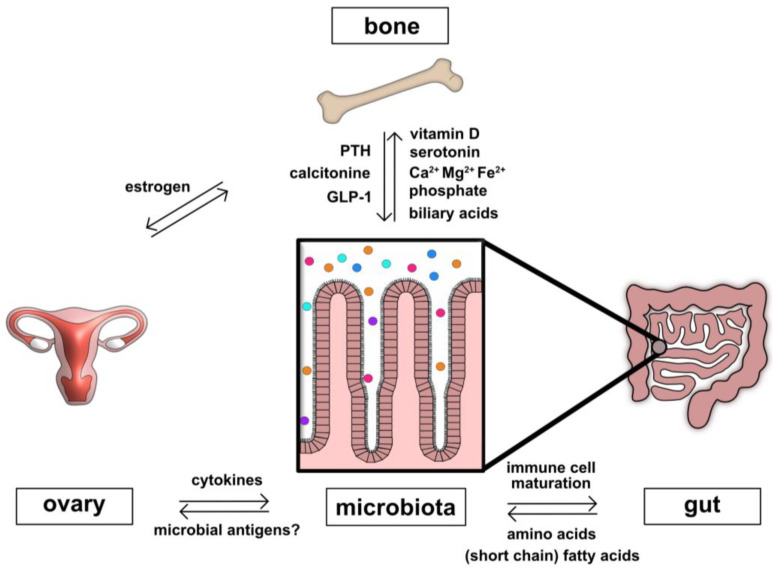
Influence of gut microbiota on physiological circuits regulating bone metabolism. Intestinal microbiota can interfere with several factors and signaling pathways regulating physiological bone homeostasis and metabolism. For example, calcitonin and PTH (hormones which exert opposite effects on mineralization) require intestinal microbiota to exert their anabolic and catabolic effects. Microbiota enzymatically activate estrogens, control cytokine production, and control immune cell differentiation, particularly Th17-Treg homeostasis. Intraluminal, microbiota-dependent amino acid and short chain fatty acid (SCFA) metabolism contributes (in part) to the regulation of these immune responses. Intestinal microbiota are crucial for the maintenance of the intestinal physiological barrier and electrolyte balance. Vitamin D is pivotal to establishing microbial diversity in the gut.

**Figure 2 ijms-24-05161-f002:**
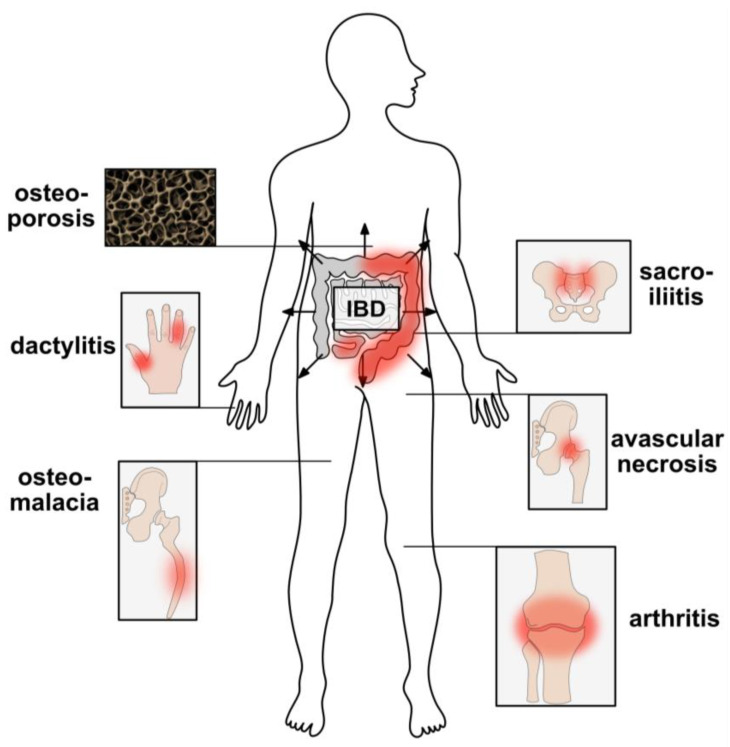
Extraintestinal disease manifestations of IBD affecting bones and joints. Involvements of the joints experienced concomitantly with IBD include arthritis, sacroiliitis and dactylitis, as well as disease manifestations affecting the complete bone, such as osteoporosis, osteomalacia and avascular necrosis.

**Figure 3 ijms-24-05161-f003:**
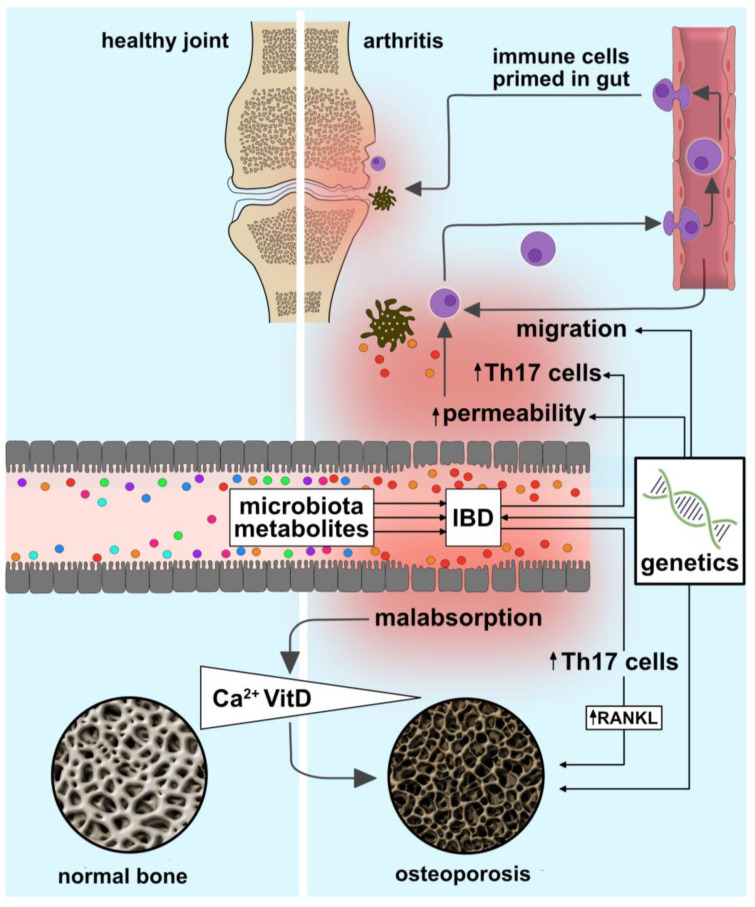
Pathogenic mechanisms underlying bone disease in IBD. IBD is a multifactorial disease driven by environmental factors and complex genetic traits. Gut microbial dysbiosis, intestinal barrier disruption, dysregulated intraluminal metabolism and aberrant immune responses characteristically accompany IBD. For example, dysbiosis-driven immune cell priming mitigates the expansion, the egress and the migration of Th17 cells from the gut into the joints, causing arthritis. Enhanced angiogenesis further promotes inflammatory cell and cytokine transport. Both the inflammation-associated disruption of the intestinal barrier and dysbiotic microbiota can lead to a malabsorption of calcium and disruptions of vitamin D metabolism, underlying the development of osteoporosis.

## Data Availability

All data are available in the presented manuscript.

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
