# Peer review of "Pivotal Role of Intestinal Microbiota and Intraluminal Metabolites for the Maintenance of Gut–Bone Physiology"

_ijms, 2023, doi:10.3390/ijms24065161_

Round 1
Reviewer 1 Report
Manuscript ID: Ijms-2216761 REVIEW REPORT
Type of manuscript: Review
Title: Pivotal role of intestinal microbiota and intraluminal metabolites for the maintenance of gut bone physiology
The authors summarized evidences on the influence of gut microbiota , metabolites and microbiota-primed immune cells in IBD and bone-related complications.
.
COMMENTS
In my opinion the paper is well written and the authors did an appreciable work.
I have not major or minor remarks
Author Response
The authors would like to thank the referee for his/her appreciation of the importance of our work.
Reviewer 2 Report
In this review article "Pivotal role of intestinal microbiota and intraluminal metabolites for the maintenance of gut bone physiology". The authors summerized the role of IBD with bone disease. This review is well designed and provided new concept in this field.
1. The references are too old. More recenlty refs (5 years) are more sutiable
2. The promising medicines should be discussed
3. The further application should be discussed.
Author Response
The authors would like to thank the referee for his/her appreciation of the importance of our work and for his/her helpful and constructive critique. Below please find a point-by-point reply to all issues raised.
All pages and lines mentioned in the point-by-point reply refer to the revised manuscript version in which the respective changes are highlighted in tracker.
- The references are too old. More recenetly refs (5 years) are more suitable
Almost 2/3 of the references cited in our review article have been published within the last five years.
- The promising medicines should be discussed
We discuss FMT as potential therapeutic option to cure next to the intestinal symptoms also extra-intestinal manifestations of IBD and suggest in lines 517-519 that „ it would be intriguing to treat IBD by changing gut microbiota and intraluminal metabolites by simple dietary supplements.“ Next to FMTs also the application of distinct microbial metabolites such as SCFAs might be promising as well as mentioned now in addition at the end of the discussion in lines 529 and 530.
- The further application should be discussed.
FMT improve Th1/Th2 and Th17/Treg imbalance that affect also „musculoskeletal manifestations of IBD“ as stated in line 525. To correct this imbalance the application of distinct microbial metabolites and/or intestinal microbiota might be a promising tool.
- Extensive editing of English language and style required
We have carefully gone through the manuscript and have hopefully corrected all misspellings and confusing expressions/statements. Could you please specify in which parts English editing is required.
Reviewer 3 Report
The manuscript titled "Pivotal Role of Intestinal Microbiota and Intraluminal Metabolites for the Maintenance of Gut-Bone Physiology" by Niklas Grüner et al. is an interesting contribution to the field of bone health. The article is well-written and presents the current state of knowledge, as well as the aims of the study, effectively.
I recommend this manuscript for publication, but I do have a few minor comments and questions:
Figure 1: The schematic diagram of the internal small intestinal epithelial cells in the cross-sectional image of the intestine should be replaced with the villus-crypt structure diagram.
Line 89: Traditionally, calcium and phosphorus are absorbed only in the small intestine. It may be better to replace "the intestinal barrier" with "the small intestinal barrier."
Line 338: Acetate, butyrate, and propionate are short-chain fatty acids (SCFAs). The author may want to clarify by using the term "short-chain fatty acids (SCFAs), including acetate, butyrate, and propionate."
Line 338-339 and Line 345: Line 345 mentions that SCFAs do not affect bone formation, while Line 338-339 suggests that SCFAs enhance alkaline phosphatase activity, which is a marker of bone formation. This is contradictory. In fact, the protective effects of SCFAs are mainly associated with the inhibition of osteoclast differentiation and maturation. It is better to rewrite this part.
Could the author provide some examples of microbial metabolites that directly affect the differentiation and maturation of osteoblasts and osteoclasts, such as SCFAs?
Author Response
The authors would like to thank the referee for his/her appreciation of the importance of our work and for his/her helpful and constructive critique. Below please find a point-by-point reply to all issues raised.
All pages and lines mentioned in the point-by-point reply refer to the revised manuscript version in which the respective changes are highlighted in tracker.
I recommend this manuscript for publication, but I do have a few minor comments and questions:
Figure 1: The schematic diagram of the internal small intestinal epithelial cells in the cross-sectional image of the intestine should be replaced with the villus-crypt structure diagram.
Figure 1 has been corrected accordingly.
Line 89: Traditionally, calcium and phosphorus are absorbed only in the small intestine. It may be better to replace "the intestinal barrier" with "the small intestinal barrier."
This has been corrected accordingly in line 89.
Line 338: Acetate, butyrate, and propionate are short-chain fatty acids (SCFAs). The author may want to clarify by using the term "short-chain fatty acids (SCFAs), including acetate, butyrate, and propionate."
Yes, that is correct. Acetate, butyrate, and propionate are short-chain fatty acids (SCFAs). We modified the sentence in lines 337-339 that reads how: „One of the microbiota-derived metabolites that potentially can influence bone homeostasis are the short chain fatty acids (SCFAs), mainly consisting of acetate, butyrate and propionate.“
Line 338-339 and Line 345: Line 345 mentions that SCFAs do not affect bone formation, while Line 338-339 suggests that SCFAs enhance alkaline phosphatase activity, which is a marker of bone formation. This is contradictory. In fact, the protective effects of SCFAs are mainly associated with the inhibition of osteoclast differentiation and maturation. It is better to rewrite this part.
We slightly modified the respective paragraph spannimng lines 337 - 346 in order to eliminate contradictory messages. Its final sentence in lines 344-346 reads now: „The protective effects of SCFAs on bone mass are primarily associated with inhibition of osteoclast differentiation and bone resorption, while bone formation is less affected“
Could the author provide some examples of microbial metabolites that directly affect the differentiation and maturation of osteoblasts and osteoclasts, such as SCFAs?
In line 140-142 it says that„short chain fatty acids (SCFAs) released by microbiota upon carbohydrate digestion are further important modulators of bone physiology.“ Two of the cited references are described again in greater detail in lines 337-346. One of them is an in vitro study that shows that acetate and propionate directly upregulate osteoblastic differentiation. The other study showed that the protective effects of SCFAs on bone mass were associated with an inhibition of osteoclast differentiation and bone resorption in vitro and in vivo.